# Exploring the Toxic Effects of ZEA on IPEC-J2 Cells from the Inflammatory Response and Apoptosis

**DOI:** 10.3390/ani13172731

**Published:** 2023-08-28

**Authors:** Haoyue Guan, Wenxue Ma, Qiong Wu, Jingzeng Cai, Ziwei Zhang

**Affiliations:** 1College of Veterinary Medicine, Northeast Agricultural University, Harbin 150030, China; guanhaoyue08@163.com (H.G.); mawenxue2023@163.com (W.M.); caijingzeng@neau.edu.cn (J.C.); 2College of Animal Science and Veterinary Medicine, Sichuan Agricultural University, Chengdu 611130, China; 3Animal Science and Technology College, Beijing University of Agriculture, Beijing 100096, China

**Keywords:** zearalenone, IPEC-J2 cells, oxidative stress, apoptosis, inflammatory factors

## Abstract

**Simple Summary:**

Zearalenone is a fungus that is often found in the ingredients of many animal feeds. Swine, as the animal most sensitive to zearalenone, have more severe symptoms of infection and therefore can cause great economic losses. The intestine is the first line of defence of the organism, so we constructed a model of zearalenone intoxication in swine small intestinal epithelial cell lines and performed various tests. Finally, we found that zearalenone can trigger oxidative stress, induce inflammatory response and apoptosis in porcine small intestinal cells, and finally trigger damage to porcine small intestinal tissues. We believe that the study of the effects of zearalenone on the intestinal tract of swine is of great significance for the prevention of zearalenone infection in swine, as well as for the treatment of infection.

**Abstract:**

Zearalenone (ZEA) is the most common fungal toxin contaminating livestock and poultry feeding, especially in pigs, causing severe toxic effects and economic losses. However, the mechanism of ZEA damage to the intestine is unknown. We constructed an in vitro model of ZEA toxicity in a porcine small intestinal epithelial cell (IPEC-J2) line. ZEA causes severe oxidative stress in porcine small intestine cells, such as the production of ROS and a significant decrease in the levels of antioxidant enzymes GSH, CAT, SOD, and T-AOC. ZEA also caused apoptosis in porcine small intestine cells, resulting in a significant reduction in protein and/or mRNA expression of apoptosis-related pathway factors such as P53, caspase 3, caspase 9, Bax, and Cyt-c, which in turn caused a significant decrease in protein and/or mRNA expression of inflammatory-related factors such as IL-1β, IL-2, Cox-2, NF-κD, NLRP3, IL-6, and IL -18, which in turn caused a significant increase in protein and/or mRNA expression levels. The final results suggest that ZEA can cause a severe toxic response in porcine small intestine cells, with oxidative stress, apoptotic cell death and inflammatory damage.

## 1. Introduction

ZEA is a fungal toxin that can be produced by several fungi, including *Fusarium graminearum*, *Fusarium culmorum*, *Fusarium cerealis*, *Fusarium equisetiand*, and *Fusarium semitectum* (Richard, 2007; Bennett and Klich, 2003) [1,2]. ZEA is commonly found in grains such as wheat, barley, maize, sorghum, rye, and maize silage, which are essential components of animal feeds (Zinedine et al., 2007) [3]. Swine are the most sensitive animals to ZEA, and when infected, they can suffer from severe symptoms such as labial prolapse and rectal prolapse, especially in pre-pubertal sows (BJ Blaney et al., 1984) [4]. Due to its affinity for the oestrogen receptor, it has a more pronounced effect primarily on the reproductive organs (Sundlof and Strickland, 1986; Curtui et al., 2001) [5,6]. Zearalenone and its metabolites are immunotoxic and, furthermore, can alter the viability and proliferation of the body’s immune cells, the cell cycle, and the functions of immune cells such as the inflammatory response and their ability to synthesise reactive molecules [7]. Because the metabolism of ZEA occurs mainly in enterocytes, peripheral blood erythrocytes and hepatocytes, the entero-hepatic cycle in pigs prolongs the exposure of ZEA in the digestive tract, so the effect of zearalenone on the pig intestine is also essential [8]. Although low doses of ZEA have little or no impact on the epithelial cells covering the small intestinal mucosa, these toxins significantly affect the intestinal immune system. They may lead to the development of subclinical inflammation and allergy [9].

With the increasing research on ZEA, the oxidative stress induced by ZEA in the organism is of broad interest. In general, the main by-product of biological metabolism is reactive oxygen species (ROS), which are composed of many reactive molecules derived from O_2_^−^ or H_2_O_2_ [10]. A moderate amount of ROS can act as a second messenger to regulate cell signalling pathways. However, excessive ROS can lead to the development and progression of several diseases, such as the better-known cancers [11]. ZEA can induce liver damage, subsequent development of hepatocellular carcinoma [12], and alterations in several enzymatic liver function parameters in sows. Previous reports have clearly indicated that ZEA induces lipid peroxidation [13], which can lead to the induction of oxidative DNA damage [14] as well as apoptosis [13], so this considerable cytotoxicity and genotoxicity may be related to the intracellular oxidative stress generated by ZEA. There is a close link between inflammation and oxidative stress, and the pathways that produce inflammatory mediators are induced by oxidative stress. In the current studies, there are fewer studies on the toxic effects of ZEA on the swine intestine, even though pigs are among the most sensitive animals to ZEA and its metabolites.

Therefore, we constructed a model of ZEA toxicity in IPEC-J2 cell lines to elucidate the toxic effects of ZEA on IPEC-J2 cells through the detection of oxidative stress levels, inflammatory factors, and apoptosis-related genes.

## 2. Materials and Methods

### 2.1. Cell Culture and Grouping

IPEC-J2 cells used in this experiment were provided from the College of Animal Science, Northeast Agricultural University. The cells were first cultured using Eagle medium DMEM high glucose (GIBCO, NY, USA) medium as a liquid environment. The DMEM high glucose medium, FBS and penicillin-streptomycin were then sterilised using a 0.22 mm microporous filter. Cells were fed once a day and passaged every 2 to 3 days until the cell density reached 70% to 80%. At the time of passaging, the culture supernatant was discarded and the cells were rinsed three times with PBS. Add 1 mL of digestion solution (0.25% EDTA) to the culture flask and place it in the incubator at 37 °C for 5 min. Observe the cell digestion under the microscope, and if most of the cells become rounded and detached, quickly take them back to the operating table, and terminate the digestion by tapping the flask a few times and then adding 5 mL of complete culture medium containing 10% serum. Blow the cells gently and aspirate them after complete detachment. Centrifuge the cells at 1000 r/min for 5 min, discard the supernatant, add 1~2 mL of culture medium, and then blow them well. The cell suspension was divided into new bottles containing 4~5 mL of culture medium at a ratio of 1:2. Finally, IPEC-J2 cells were treated with ZEN (20 μg/mL) (Sun Het al., 2021) [15]. All drugs were dissolved in dimethyl sulfoxide (DMSO) and the same concentration of DMSO was added to the control group. Concurrent trials have demonstrated that DMSO at concentrations below 0.02% is not toxic to IPEC-J2 cells, and none of the drugs used in the trials contained more than 0.02% DMSO.

### 2.2. Measurement of Oxidative Stress and Antioxidant Indicators

The assay kits assayed several important oxidative stress and antioxidant indicators to be tested in this experiment. Superoxide dismutase (SOD, Nanjing Jiancheng Bioengineering Institute, Nanjing, China) was determined by the xanthine oxidase method; total antioxidant capacity (T-AOC, Nanjing Jiancheng Bioengineering Institute, Nanjing, China) and malondialdehyde (MDA, Nanjing Jiancheng Bioengineering Institute, Nanjing, China) were determined by the colorimetric method; reduced glutathione (GSH, Geruisi Bio, Suzhou, China) was determined by the spectroscopic method; determination of Certified Accounting Technician (CAT, Nanjing Jiancheng Bioengineering Institute, Nanjing, China) using visible light emission.

### 2.3. ROS Activity Assay for IPEC-J2 Cells

This experiment measured ROS activity using a ROS assay kit (Nanjing Jiancheng Bioengineering Institute, Nanjing, China). First, 10 μmol/L 2,7-dichloro-dihydro-fluorescein diacetate (DCFH-DA) was added to the culture medium containing the cell samples to be tested and incubated in a constant temperature incubator (37 °C) for 45 min. The medium was discarded and the cells were washed 3 times using PBS (37 °C preheat). Finally, the cells were collected and the activity of ROS at excitation wavelength 500 ± 15 nm and emission wavelength 530 ± 20 nm was assayed. Finally, IPEC-J2 cells were observed using fluorescence microscopy.

### 2.4. Hoechst Staining

According to the manufacturer’s instructions, the Hoechst 33,343 staining kit was used to detect cell death. Cells were treated for 24 h, washed twice with PBS and stained with the Hoechst 33,343 staining kit at 4 °C for 5 min in a dark environment. Finally, the cells were observed and photographed using a fluorescent microscope.

### 2.5. Apoptotic Cell Death Assay

Acridine Orange/Ethidium Bromide (AO/EB) assay was used to detect living and apoptotic cells.AO can penetrate cells with intact cell membranes and embedded in the nuclear DNA, which emits bright green fluorescence. EB can only penetrate cells with damaged cell membranes and embedded in the nuclear DNA, which emits reddish-orange fluorescence. Apoptotic cells show enhanced staining, brighter fluorescence, and uniform circular or condensed, clumped structures. The nuclei of non-apoptotic cells show structure-like features with varying shades of fluorescence. Therefore, AO is usually used for double staining with EB to distinguish normal cells from apoptotic cells. IPEC-J2 cells were first grown in 12-well plates, washed with PBS and stained with AO/EB for 5 min. Finally, IPEC-J2 cells were observed and analysed under fluorescence microscope.

### 2.6. Detection of mRNA Expression of Apoptosis and Inflammatory Factor-Related Pathways

Total RNA was first isolated from small intestinal tissues and intercellular spaces using Trizol reagent (Invitrogen, Shanghai, China), and then the dried RNA particles were resuspended in 50 μL of diethyl pyrocarbonate-treated water. The concentration and purity of total RNA were then determined using a spectrophotometer. The cDNA was synthesised from 5 μg of total RNA using oligonucleotide primers and Superscript II reverse transcriptase according to the instructions provided by the manufacturer (Promega, Beijing, China), and the cDNA was stored at −80 °C. The Reverse Transcription-Polymerase Chain Reaction (RT-PCR) method to detect apoptosis and inflammation-associated genes. such as NLRP3, Caspase-3, Caspase-9, interleukin 18 (IL-18), and interleukin 1-beta (IL-1β), were detected.

### 2.7. Protein Expression Detection of Apoptosis and Inflammatory Factor-Related Pathways

IPEC-J2 was washed three times and then the total protein was extracted using a 100:1 ratio of RIPA lysate (Biosharp, Beijing, China), and the protein was blotted with phenylmethylsulphonyl fluoride (PMSF) (100 mmol/L). Prepared sodium dodecyl sulphate–polyacrylamide gel electrophoresis (SDS-PAGE) gels at 12% and/or 10% concentrations were separated from the proteins by SDS–polyacrylamide gel electrophoresis and transferred to nitrocellulose (NC) membranes at a constant current of 200 mA. The transferred membranes were placed in an incubator with TBST and closed with 5% bovine serum albumin (BSA) for 2 h, and incubated overnight with diluted primary antibodies against Bcl-2 (1:500, polyclonal antibody produced in our laboratory), Bax (1:500, polyclonal antibody produced in our laboratory), Bad (1:500, poly clone antibodies), caspase 9 (1:500, polyclonal antibody produced in our laboratory), P53, COX-2 (1:1000, Santa Cruz, CA, USA), IL-1β (1:1500, Wanleibio, Shenyang, China), IL-2 (1:1500, Wanleibio, China), NF-κB (1:1000, Cell Signaling Technology, Inc., Danvers, MA, USA). Subsequently, the membrane was incubated with an anti-rabbit immunoglobulin G (IgG) antibody for 1 h at 37 °C and washed 3 times with TBST for 10–15 min each time. β-Actin content was used as an internal reference. Finally, bands with the chemiluminescence imaging system (Azure Biosystems C300, Azure Biosystems Inc., Dublin, CA, USA) were detected with the ECL kit (Biosharp, Beijing, China).

### 2.8. Statistical Analysis

Each group consisted of six separate observation replicates (*n* = 6), and two parallel experiments were formed to ensure the accuracy of the experimental data. All data were expressed as mean ± standard deviation (SD), Data were statistically analysed using GraphPad Prism v8.0 software and compared using a *t*-test analysis of variance to determine differences between the control and ZEA poisoning groups. An asterisk (*) denotes a significant difference from the corresponding control (*p* < 0.05).

## 3. Results

### 3.1. ZEA-Induced Changes in Oxidative Stress Parameters in IPEC-J2 Cells

To assess the effect of ZEA on oxidative stress parameters in IPEC-J2 cells, the activities of the antioxidant enzymes GSH, CAT, SOD, and T-AOC and the levels of peroxymalondialdehyde (MDA) were measured using the kit. As shown in Figure 1, the antioxidant enzymes GSH activity decreased by 12.8%, CAT activity decreased by 61.3%, SOD activity decreased by 30.6% and T-AOC activity decreased by 7.9% in the ZEA group compared to the control group (*p* < 0.05). In contrast, the level of the peroxidation product MDA was significantly increased, with a 30.2% increase in MDA content compared to the control group (*p* < 0.05).

### 3.2. Oxidative Stress Levels in IPEC-J2 Cells under the Influence of ZEA

To elucidate the role of ZEA in mitochondrial function, we examined the intracellular ROS levels in IPEC-J2 cells. DCFH-DA is a fluorescent probe for ROS. Finally, intracellular ROS production was observed by fluorescence microscopy. The number of ROS-stained positive cells in the ZEA group was 14.6-fold higher than in the control group (Figure 2). It indicated that ZEA enhanced ROS production in IPEC-J2 cells. This result is the best corroboration that ZEA can cause oxidative stress in IPEC-J2 cells.

### 3.3. Cell Death Assays of IPEC-J2 Cells

In this experiment, AO/EB staining and Hoechst staining were used to assess the apoptosis of IPEC-J2 cells. Figure 3A shows the results of Hoechst staining of porcine small intestine tissue. Hoechst’s stain penetrates the intact cell membrane and causes the DNA in the nucleus of normal cells to fluoresce blue, whereas in apoptotic cells the DNA in the nucleus undergoes breakage and coalescence leading to a change in the distribution of the fluorescent dye, which results in enhanced staining and brighter fluorescence. Although some apoptotic cells were also seen in the control group, as the number of days in culture increased, the number of apoptotic cells in the ZEA group was significantly greater. Figure 3B shows the results of AO/EB staining of pig small intestine tissue. The green fluorescence is normal cells, the red fluorescence is dead cells, and the superimposed orange colour is apoptotic cells. Compared to the control group, the ZEA group showed a significant decrease in viable cells and a significant increase in apoptotic cells, suggesting that ZEA may induce apoptosis in IPEC-J2 cells.

### 3.4. Effect of ZEA on Apoptosis and mRNA Expression of Inflammatory Response-Relb Gated Proteins in Porcine IPEC-J2 Cells

We performed RT-PCR on the relevant genes to elucidate the ZEA-induced inflammatory response and apoptosis in IPEC-J2 cells. The abundance of mRNA expression in IPEC-J2 cells after ZEA intoxication is shown in Figure 4 and Figure 5. The results of mRNA expression of inflammatory response-related genes are shown in Figure 4. mRNA expression of NF-κB, NLRP3, IL-1β, IL-6, IL-18, IL-2, and Cox-2 was substantially upregulated in the ZEA group compared to the control group (*p* < 0.05). The Cox-2 gene was not expressed in most tissue cells under normal physiological conditions. In contrast, the expression of the Cox-2 gene in the ZEA group was nearly 4.3-fold higher than that in the control group. This suggests that ZEA produced inflammatory stimulation in IPEC-J2 cells. The results of mRNA expression of apoptosis-related genes are shown in Figure 5. Compared with the control group, the mRNA expression of P53, caspase 3, caspase 9, Bax, and Cyt-c were significantly upregulated, and the mRNA expression of Bcl-2 was reduced considerably by a factor of two (*p* < 0.05).

### 3.5. Effect of ZEA on the Expression of Proteins Associated with IPEC-J2 Cells

The protein expression abundance of inflammation-related genes and apoptosis-related genes detected by WB analysis in porcine small intestine tissue is shown in Figure 6. As seen in the results of inflammation-related protein expression (Figure 6A, Appendix A), the protein expression of IL-1β, IL-2, Cox-2, and NF-κB was significantly increased in the ZEA-intoxicated group compared to the control group (*p* < 0.05), and the protein overexpression of these cytokines may lead to an inflammatory response of IPEC-J2 cells to ZEA. In the protein expression results of apoptosis-related genes (Figure 6B), the protein expression of P53, caspase9 and Bax was significantly increased, and the protein expression of Bcl-2 was decreased in the ZEA intoxication group compared to the control group (*p* < 0.05). This represents a significant increase in the Bax/Bcl-2 ratio in the ZEA-intoxicated group, and the elevated Bax/Bcl-2 ratio may suggest that ZEA directs apoptosis. The above results show that ZEA can induce an inflammatory response as well as apoptosis in porcine small intestinal tissue.

## 4. Discussion

Among the fungal toxins identified, ZEA has been recorded as the most common toxin in the northern hemisphere over a wide range of temperatures [16,17]. ZEA is frequently found in different cereals such as wheat, barley, and wheat, maize, sorghum, rye, and maize silage, all of which are components of many animal feeds [18,19]. According to statistics, ZEA creates a global loss of millions of dollars annually [20]. It is of concern to all that when ZEA enters the organism. The intestine must first absorb it, the first physical barrier against foreign substances. As mycotoxin-contaminated feed is absorbed, the intestine and its epithelial cell layer will be exposed to high concentrations of the toxin, which will undoubtedly affect intestinal health. Because swine are considered the most sensitive animals to ZEA, it can have severe consequences if it causes widespread poisoning.

In the natural environment, the concentration of ZEA varies in different specific substances and regions. In previous reports, it was shown that ZEA concentrations ranged from 86 ng/g_dryweight(dw)_ to 16.7 μg/g_dw_ in wheat samples, 126 ng/g_dw_ to 13.8 μg/g in maize, and up to 7.5 ng/g_dw_ in soil [21], while toxin levels in water samples ranged from 0.5 to 4.9 ng/L [22]. In this experiment, we chose a ZEA concentration of 20 μ/mL, which is much higher than the concentration of ZEA in the natural environment. The purpose of adding a high concentration of ZEA is mainly to study the effect of ZEA on the apoptosis and inflammation of IPEC-J2 cells, which is more reasonable for theoretical studies. The present experiment is a study to determine the effects of ZEA on intestinal toxicity in swine, and future studies should be conducted at more environmentally friendly concentrations.

Oxidative stress is associated with damage to cellular structures and many diseases, such as cancer [23], and previous studies have shown that ZEA entry into the body enhances ROS formation and leads to oxidative damage [24]. ZEA-induced oxidative stress may be one of the primary mechanisms by which ZEA induces cell damage and genomic toxicity [11,25]. The current study shows that high concentrations of ZEA severely impair the antioxidant enzyme system of IPEC-J2 cells and increase the production of ROS, thereby exacerbating mitochondrial damage [26]. We also detected a significant increase in ROS in the IPEC-J2 of the ZEA-intoxicated group in our experiments. MDA is considered an excellent indicator for determining lipid peroxidation. Our results show a significant increase in MDA concentration in the ZEA-intoxicated group. O_2_ is the initial ROS produced by the mitochondrial respiratory chain, mainly by complexes I and III. It can then be quickly converted to H_2_O_2_ by SOD and reduced to water by CAT or GSH-PX [27]. SOD activity is known to be protective against the elimination of reactive free radicals, making it an important antioxidant in almost all cells exposed to oxygen [28]. GSH-PX can change toxic peroxides into non-toxic hydroxyl compounds to protect membrane structure and function. CAT is an early marker of oxidative stress. T-AOC represents the total antioxidant level in the organism consisting of various antioxidant substances and antioxidant enzymes, etc., as these antioxidant substances and antioxidant enzymes protect cells and the organism from oxidative stress damage caused by reactive oxygen radicals [29]. In our results (Figure 3), the concentrations of GSH, CAT and SOD were significantly decreased in the ZEA-poisoned group compared to the control group, indicating that ZEA poisoning enhances the formation of ROS in IPEC-J2 cells, which causes oxidative stress. The T-AOC concentrations in the ZEA-poisoned group were slightly decreased, indicating that ZEA reduces the overall antioxidant level in IPEC-J2 cells.

ZEA has been described as a suppressor and inducer of inflammation [30,31,32,33,34]. ZEA affects the synthesis of inflammatory cytokines in the liver and spleen and the expression of genes involved in inflammation [35]. ZEA and its metabolites can also interfere with the barrier function of the intestine and the ability of intestinal cells to achieve an inflammatory response [36]. Recently, researchers have demonstrated that ZEA can lead to enhanced cytokines such as IL-1β and IL-18 in IPEC-J2 cells, mouse peritoneal macrophages, and colon tissue. ZEA induces caspase-1 activation via the NLRP3 inflammatory vesicle complex, cleaving pro-IL-1 and pro-IL-18 into their biologically active forms, thereby initiating the intestinal inflammatory cascade response; the hyperenhancement of these cytokines leads to an inflammatory response of the gut to ZEA [37]. At high ZEA concentrations in vitro, alterations in immune parameters, such as inhibition of mitogenically stimulated lymphocyte proliferation and increased IL-2 production can be found [38]. The results in our experiment clearly show a significant upregulation of NLRP3, IL-1β, IL-18 and IL-2 gene expression in IPEC-J2 cells in the ZEA group, consistent with the above findings that ZEA can induce an inflammatory response and enhance the presentation of inflammatory factors in the swine small intestine. We also examined the expression of the Cox-2 protein. This inducible enzyme can be highly expressed when cells are stimulated by inflammation and is an essential determinant of inflammation-mediated cytotoxicity. Cox-2 is not defined in most tissues under physiological conditions. Still, its expression tends to increase when induced by pro-inflammatory agents such as inflammatory stimuli and injury in pathological conditions such as inflammation and tumours. Cox-2 is not only associated with inflammation but may also play an essential role in apoptosis by inhibiting the activity of regulatory genes such as Bcl-2 and caspase-3 [39]. Early studies found that the emergence of inflammation was accompanied by the emergence of apoptosis, as in vitro tests demonstrated that inflammation-related stimuli induced apoptosis [40]. Apoptosis plays an essential role in immunogenesis, during inflammation, and in the resolution of inflammatory responses. There is evidence that peripheral cells adjacent to inflammation die themselves through apoptosis, thereby increasing the damage caused by the inflammatory response [41]. Thus, ZEA also affects apoptosis. When the concept of apoptosis was first introduced, apoptosis was classified into two forms: endogenous and exogenous. The mitochondrial pathway is the most typical endogenous apoptotic pathway and is activated by the penetration of Bcl-2 family genes into the outer mitochondrial membrane, resulting in the release of mitochondrial proteins into the cytoplasm, including Cytochrome c (Cyt c) and the second generation mitochondria-derived cysteine aspartase activator (SMAC), which promotes the formation of apoptotic vesicles and activates caspase 3, caspase 7, and caspases 9, thereby synergizing cell death [42,43]. ZEA has been experimentally demonstrated to induce apoptosis and necrosis in porcine granulosa cells via caspase 3- and caspase 9-dependent mitochondrial pathways [44]. P53 functions mainly as a transcription factor, exerting its downstream functions by activating or repressing many genes that initiate one of the three main programs of cell cycle arrest, DNA repair or apoptosis [45]. Therefore, in our experiment, we examined the mRNA and protein expression of Bcl-2, Bax, P53 and caspases 9, critical genes in the apoptotic pathway, and all three of these genes, except for Bcl-2, were significantly increased in the ZEA-intoxicated group. Of these, the Bax/Bcl-2 ratio was significantly increased in the ZEA intoxication group. This suggests that ZEA promotes apoptosis and damage to IPEC-J2 cells in the intestine, as a more significant proportion of Bax/Bcl-2 directs apoptosis.

## 5. Conclusions

In conclusion, in the porcine small intestine epithelial cell line ZEA toxicity model constructed in this experiment, ZEA toxicity can induce inflammatory responses and apoptosis by enhancing the formation of ROS in the porcine small intestine epithelial cells, causing oxidative stress as well as decreasing the overall anti-oxidation level, and inducing inflammatory responses and apoptosis in the porcine small intestine by upregulating the gene expression of inflammatory and apoptosis-related cytokines, finally triggering damage to the porcine small intestine tissue. Because the intestine is the first barrier against external influences, if the small intestinal tissue is severely damaged, it can also impair overall body quality. Therefore, the toxic effects of ZEA on IPEC-J2 cells should be taken more seriously and should not be ignored.

## Figures and Tables

**Figure 1 animals-13-02731-f001:**
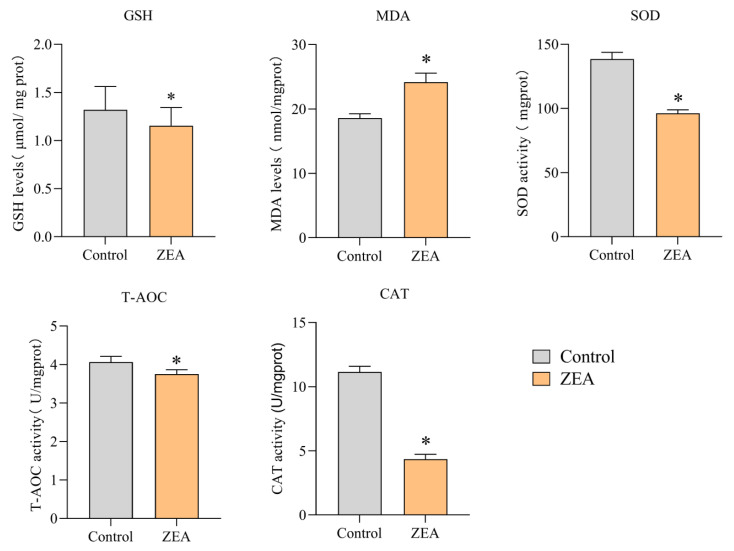
Effects of ZEA on oxidative indexes in the IPEC-J2 cells. Activities of GSH, CAT, SOD, T-AOC, and MDA in IPEC-J2 cells. * indicates significant differences from the corresponding normal values (*p* < 0.05).

**Figure 2 animals-13-02731-f002:**
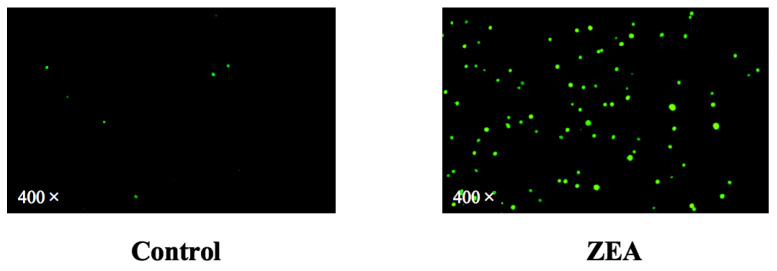
Effect of ZEA intoxication on oxidative stress in IPEC-J2. Fluorescence quantification results of ROS in IPEC-J2 by fluorescence microscopy.

**Figure 3 animals-13-02731-f003:**
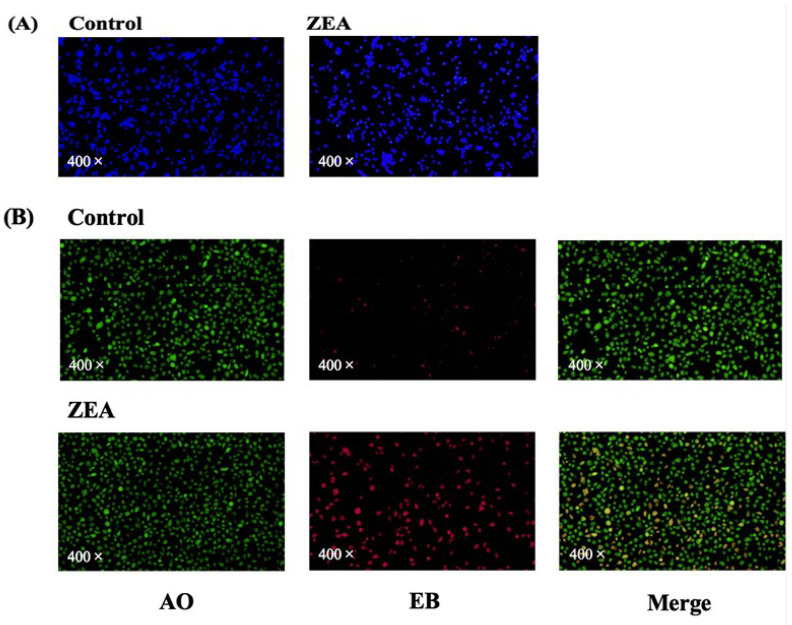
Results of apoptosis-related staining of IPEC-J2 cells. (**A**) Results of Hoechst staining observation. The nuclear DNA of normal cells shows blue fluorescence and the nuclear staining of apoptotic cells is enhanced. (**B**) Acridine orange/ethidium bromide (AO/EB) fluorescence staining. Green fluorescence is normal cells, red fluorescence is dead cells, and superimposed orange colour is apoptotic cells.

**Figure 4 animals-13-02731-f004:**
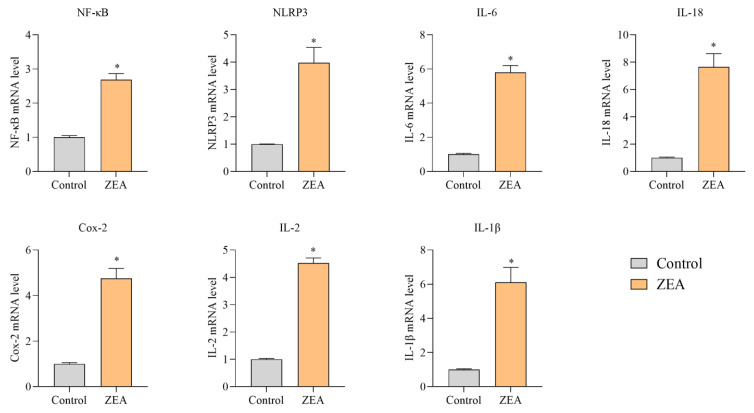
Expression levels of mRNA of inflammation-related genes in porcine intestinal tissues in the ZEA-intoxicated group and the control group. * indicates significant differences from the corresponding normal values (*p* < 0.05).

**Figure 5 animals-13-02731-f005:**
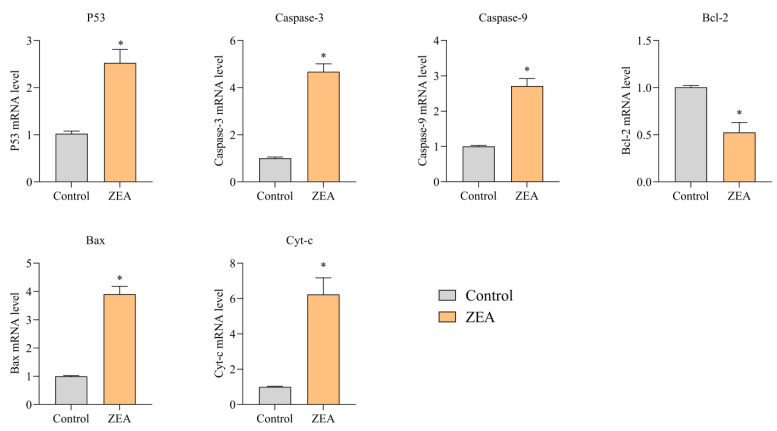
Expression levels of mRNA of apoptosis-related genes in porcine intestinal tissues in the ZEA-intoxicated group and the control group. * indicates significant differences from the corresponding normal values (*p* < 0.05).

**Figure 6 animals-13-02731-f006:**
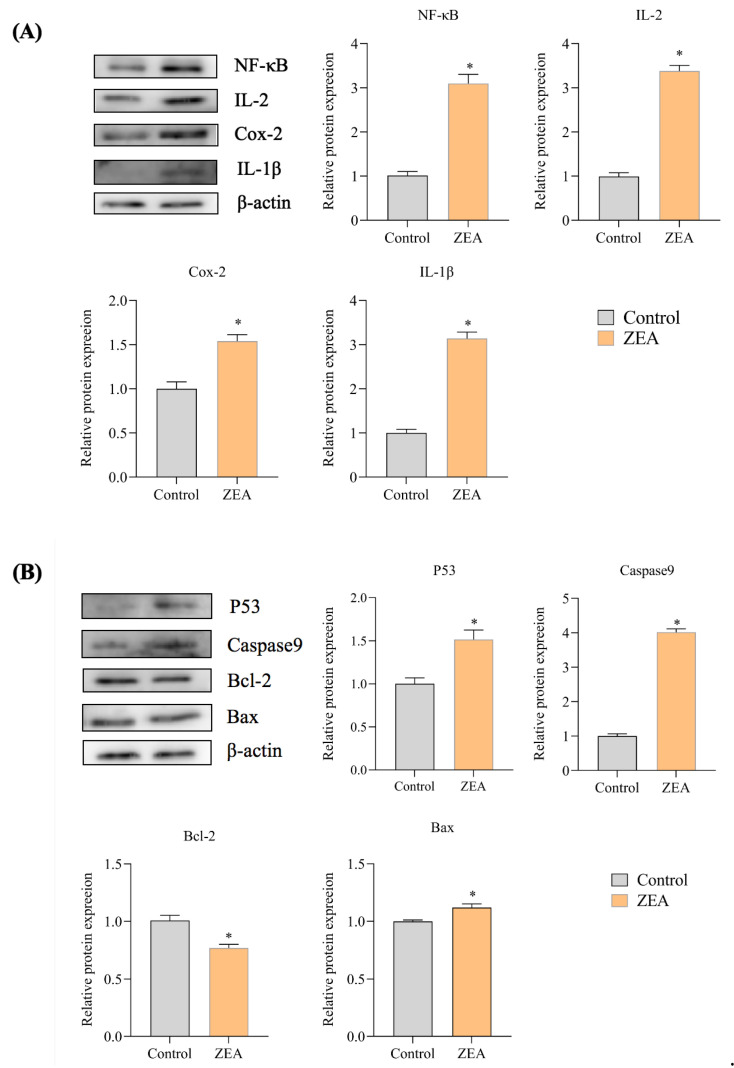
Results of apoptosis-related gene protein expression (**A**) and inflammation-related gene protein expression (**B**) in IPEC-J2 cells. * indicates significant differences from the corresponding normal values (*p* < 0.05).

## Data Availability

The data presented in this study are available on request from the corresponding author.

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
