# Peer review of "Exploring the Toxic Effects of ZEA on IPEC-J2 Cells from the Inflammatory Response and Apoptosis"

_animals, 2023, doi:10.3390/ani13172731_

Round 1

Reviewer 1 Report

In this paper, we constructed a model of zearalenone (ZEN) intoxication in a porcine small intestinal epithelial cell line, and then elucidated the toxic effects of ZEN on porcine small intestinal epithelial cells by examining the levels of oxidative stress, inflammatory factors, and apoptosis pathway-related factors. The authors concluded that ZEN intoxication could induce inflammatory responses and apoptosis in the porcine small intestine by enhancing ROS formation in porcine small intestinal epithelial cells, causing oxidative stress as well as decreasing the overall antioxidant level, and inducing damage to porcine small intestinal tissues by up-regulating the gene expression of cytokines related to inflammatory and apoptotic pathways. This is a topic of interest to researchers in related fields, but the paper needs some improvements before publication. My detailed comments are below:

(1) Lines 338-340, repeated twice “in the porcine small intestine”, making the sentence seem overly repetitive.

(2) Line 70, the phrase “despite the fact that” may be wordy. Consider changing the wording.

(3) Line 59, the phrase “a number of” may be wordy. Consider changing the wording.

(4) Regarding the various images in this article, the authors need to modify the higher resolution images.

(5) Some descriptions from the results section should be added to the discussion text.

(6) The accompanying illustrations are aesthetically pleasing and logical, but the content in the illustrations needs to be modified, e.g. fonts and sizes, positions and colours.

(7) What software was used to analyse the data statistically and what software was used for plotting?

(8) Elimination of multiple citations should be achieved by describing each citation individually, which can be done by mentioning 1 or 2 phrases in each citation to show how it differs from the others.

(9) References should pay attention to the use of punctuation and capitalisation.

(10) There are some minor problems with the references, including incomplete citations. Please double-check the references and make necessary changes. Please consider including other relevant literature and improving citations.

(11) Some of the references are outdated, and references from the last three years should be cited whenever possible.

(12) Authors should be more specific in providing the contribution of this study.

(13) The analyses of the test results should be more adequate and rational.

(14) Additional tests could be added to measure more indicators to better justify the results.

(15) The introduction section should be increased to set out the research context more clearly.

OK

Reviewer 2 Report

Please see the attached review letter.

Journal: ANIMALS Title: Exploring the toxic effects of ZEN on IPEC-J2 cells from the inflammatory response and apoptosis Type: Article Manuscript number: animals-2575332

Section: Veterinary Clinical Studies

Special Issue: Animal Poisoning Related to Pathology and Toxicology

The authors describe the construction of an in vitro model of Zearalenone toxicity, a mycotoxin produced by fungi of the genus Fusarium, using a porcine small intestinal epithelial cell (IPEC-J2) line and the identification of toxicity mechanisms through increased oxidative stress, markers of inflammation and apoptosis-related genes in porcine cells. The research proposal is in line with the section and theme addressed by the special issue.

The authors use a concentration of ZEN for cell exposure of 20 μ/ml, based on the study by Sun Het al., 2021, with the same cell line. However, it would be convenient to discuss how this dose was determined, results of pilot experiments and a parallel between this dose that simulates in vitro intoxication with the amount of toxin estimated for natural intoxications. As in vitro studies tend to use high (or low) doses that would never be seen in natural infections, this discussion or information is needed.

Still in this regard, it would also be interesting to test ZEN toxin at different concentrations to report (or not) a dose-dependent effect of toxin exposure, if this has not yet been done.

For an even more robust evaluation of the effect of the toxin on the oxidative balance, I suggest the comparison with a new experimental group formed by the addition of ZEN + antioxidant to the medium. The investigation of the effects in this last group would only be for methods that concern oxidative stress, but if possible, for all other methods. According to manuscript data, by adding an antioxidant to the culture, ZEN-induced oxidative stress would be attenuated, and a reduction of the evaluated parameters would be observed.

Minor comments:

• The parameters described by the abbreviation “CAT” used in the results, especially in section 3.1 and figure 1, are not described in methods (2.2) or elsewhere in the manuscript. This is probably about the assessment of catalase activity, but it should be added correctly in the description of the methods.

• In all figures and graphs the mycotoxin treated group is described as “ZEN” in the legend and “ZEA” in the figure.

• In item 3.3, regarding the staining mechanism, it is worth mentioning that the apoptosis cell demonstrates nuclear condensation and DNA fragmentation, and that is why apoptotic cells have enhanced nuclear staining and brighter fluorescence.

• It would be interesting to write the magnification of the photomicrographs in the figure description as well, the white bar of the images is ineligible.

.
